# Biomechanical comparison of various bone reduction forceps in interfragmentary compression and area of compression in an experimental model of canine lateral humeral condylar fractures

**Madison Baskette**[ID]*, **Cassio Ricardo Auada Ferrigno**

Department of Small Animal Clinical Science, College of Veterinary Medicine, University of Tennessee, Knoxville, Tennessee, United States of America

☯ These authors contributed equally to this work.
\* madibaskette@gmail.com

## Abstract

### Objective

To compare contact area and interfragmentary compression generated by Vulsellum forceps, Patellar forceps, Kyon FineTouch forceps, point-to-point forceps with soft-locking mechanism, and point-to-point forceps with speed-locking mechanism in simulated lateral humeral condylar fractures in canine cadavers.

### Study design

*Ex-vivo* biomechanical study.

### Materials and methods

Seven cadaveric canine humeri with simulated lateral humeral condylar fractures were used in this study. A stress-sensitive film was placed at the fracture gap and five different bone reduction forceps were used to reduce the fractures to their maximum pressure before failure occurred. The compression and interfragmentary compression area were recorded during the entire compression interval and compared after the pressure had reached a plateau.

### Results

Patellar forceps generated the highest interfragmentary compression, followed by Vulsellum forceps. Compression generated by both the Patellar and Vulsellum forceps were significantly higher than point-to-point forceps with soft-lock, point-to-point forceps with speed lock, and Kyon forceps (P = 0.0008, 0.0084). No statistically significant difference was observed in the areas of compression among all forceps types.

provided the original author and source are credited.

**Data availability statement:** Data is available from 10.6084/m9.figshare.28255967.

**Funding:** The author(s) received no specific funding for this work.

**Competing interests:** The authors have declared that no competing interests exist.

## Conclusion

Patellar and Vulsellum forceps generate a greater interfragmentary compression compared to Kyon FineTouch forceps and point-to-point forceps with both speed and soft-locking mechanisms in this experimental lateral condylar fracture model.

## Introduction

Lateral humeral condylar fractures are commonly diagnosed and repaired fractures in small animal medicine. Nearly half of all humeral fractures in canine patients involve the humeral condyle with the lateral condyle predominantly affected [1–3]. This results from specific anatomic characteristics of the distal humerus including a thinner lateral ramus and the supratrochlear foramen [3]. Furthermore, articulation with the radial head results in the greater vertical force applied to the lateral aspect of the condyle in low-energy situations (i.e., landing from a jump), resulting in a predisposition for fracture of the lateral aspect in certain patients [3]. These fractures most commonly occur in skeletally immature dogs, typically resulting in a Salter-Harris type IV fractures [1,4,5]. However, trauma or underlying orthopedic disease, such as humeral intracondylar fissure (HIF) [6,7], can also lead to fracture of the lateral humeral condyle in older patients.

Due to the articular nature of these fractures, rigid fixation and anatomical reduction are absolutely imperative [1,3,8,9]. Similarly, it is crucial to achieve interfragmentary compression to promote primary bone healing and decrease bony callous formation within the cubital joint [5,8]. Several methods of fixation have been described for lateral condylar fractures including lag screws, positional screws with and without pre-loading, self-compressing pins, and Kirschner wires [1,3,6,8–13]. Although no definitive recommendation exists, the literature most commonly describes placement of a transcondylar lag screw to generate compression at the fracture site [3]. A recent study comparing lag screws to pre-loaded positional screws concluded that positional screws provided greater interfragmentary compression than lag screws and tolerated greater torque during placement [14]. This data is contradicted by a more recent study concluding that lag screws generated both more interfragmentary compression and a larger area of compression than positional screws [15]. However, both groups in the more recent study used bone reduction forceps to hold the fracture in place during fixation [15].

While several studies have compared compression generated by lag and positional screws, studies evaluating interfragmentary compression achieved by bone reduction forceps are lacking despite frequent use of these forceps in fracture repair. Moreover, differences in locking mechanism or forcep tip could influence reduction ability and pressure at the fracture site.

Several existing instruments rely on different locking mechanisms of either the ratchet type, spin-lock, or friction type. Ratchets, generally considered to be more user-friendly, retain the desired position without assistance. However, the distance between the teeth of the ratchet can dictate the amount of compression at the fracture site, which can lead to displacement of the forceps or fracture of the bone fragment. In comparison, spin-lock forceps are fixed by the tightening of a nut, which gives the user more control of the compression of the fragments. However, these instruments require both hands, typically requiring assistance to hold fragments in reduction while compressive force is being applied. Additionally, a number of reduction forceps (i.e., Patellar, Vulsellum) have been developed with specialized tips for better purchase of tissue. These variations in tips may influence the distribution of forces on the cortices, thus leading to differences in interfragmentary compression.

The aim of this study was to compare interfragmentary compression generated by several bone holding forceps when used as a preloading device in distal humeral lateral condylar fractures. To achieve this, a simulated lateral condylar fracture was developed using canine cadaveric humeri. Our null hypotheses were that interfragmentary compression and area of compression would be of no significant difference between various bone reduction forceps.

## Materials and methods

### Fracture model

Skeletally mature humeri of dogs weighing between 8 and 16 kg were obtained from donated cadavers euthanized for reasons unrelated to the current study. Humeri were dissected clear of soft tissue and muscle attachments. Those without gross orthopedic and anatomic abnormalities were included in the study, resulting in the inclusion of 7 humeri. A lateral condylar fracture was simulated by performing an osteotomy on the midline of the humeral condyle with a sagittal saw (Colibri, Synthes) from distal to proximal. A second osteotomy was then performed at 45 degrees to the humeral shaft axis from the lateral epicondyle to the supratrochlear foramen to simulate the fracture through the lateral ramus (Fig 1).

### Interfragmentary compression

Contact area and interfragmentary compression were measured using a Tekscan pressure transducer (Tekscan Inc., South Boston, Massachusetts) and a wire film stress sensor (model 4041). Calibration was achieved through placement of a 10 kg weight on the wire film sensor while resting on a level surface. The pressure sensor was then set to display pressure in kPa, as well as to show the maximum differences among the pixels measuring the pressure.

Following calibration, the wire film stress sensor was inserted into the fracture gap between the humeral condyles and centered over the interfragmentary interface (Fig 2). The sensor was oriented perpendicular to the humeral shaft with excess sensor area on all four sides of interfragmentary interface, ensuring the entire area of compression would be captured by the sensor in each trial (Fig 2). The fracture was anatomically reduced via digital manipulation prior to application of bone reduction forceps. Real-time recordings were then performed, beginning with the minimal force required to hold the fracture fragment in reduction with the tips of the forceps at or near the epicondyles. The bone forceps were then engaged until the maximum interfragmentary compression prior to bone fragment fracture, forcep disengagement, or loss of reduction was achieved. Compression using bone forceps was performed by the same orthopedic clinical faculty in each trial with the velocity to achieve interfragmentary compression similar to clinical settings.

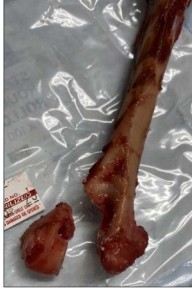

**Fig 1. This image shows a canine cadaveric humerus following two osteotomies (first on midline of the humeral condyle and second at 45 degrees to the humeral shaft axis through the lateral ramus) to simulate a lateral condylar fracture.**

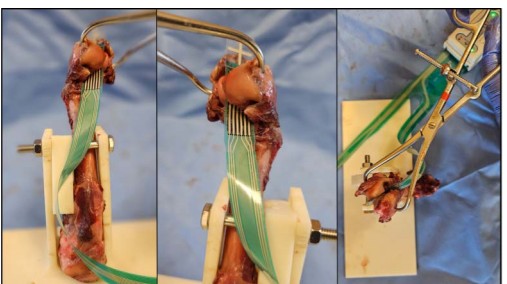

**Fig 2. These images shows a sample humerus dissected of muscle attachments with wire film stress sensor (model 4041) placed in the fracture gap.** In this image, fracture is being compressed with speed-locking point-to-points bone forceps placed at the lateral and medial epicondyles.

## Bone reduction forcep groups

The compression at the fracture site was tested using five experimental groups (Figs 3 and 4): group A (Kyon FineTouch forceps), group B (patellar forceps, DePuy Synthes [DPS] 399.085), group C (vulsellum forceps with ratchet locking), group D (point-to-point forceps with soft-locking mechanism, DPS 399.086 Reduction forceps with points, soft lock), and group E (point-to-point forceps with speed-locking mechanism (DPS 399.770 Reductions forceps).

Each forcep was applied to individual humeri three times and data was collected over the entire period of compression.

## Data collection

The area of compression (cm²), compressive force (kg), and peak pressure (kPa) were recorded over the entire period of compression for each trial. Peak pressures and area of

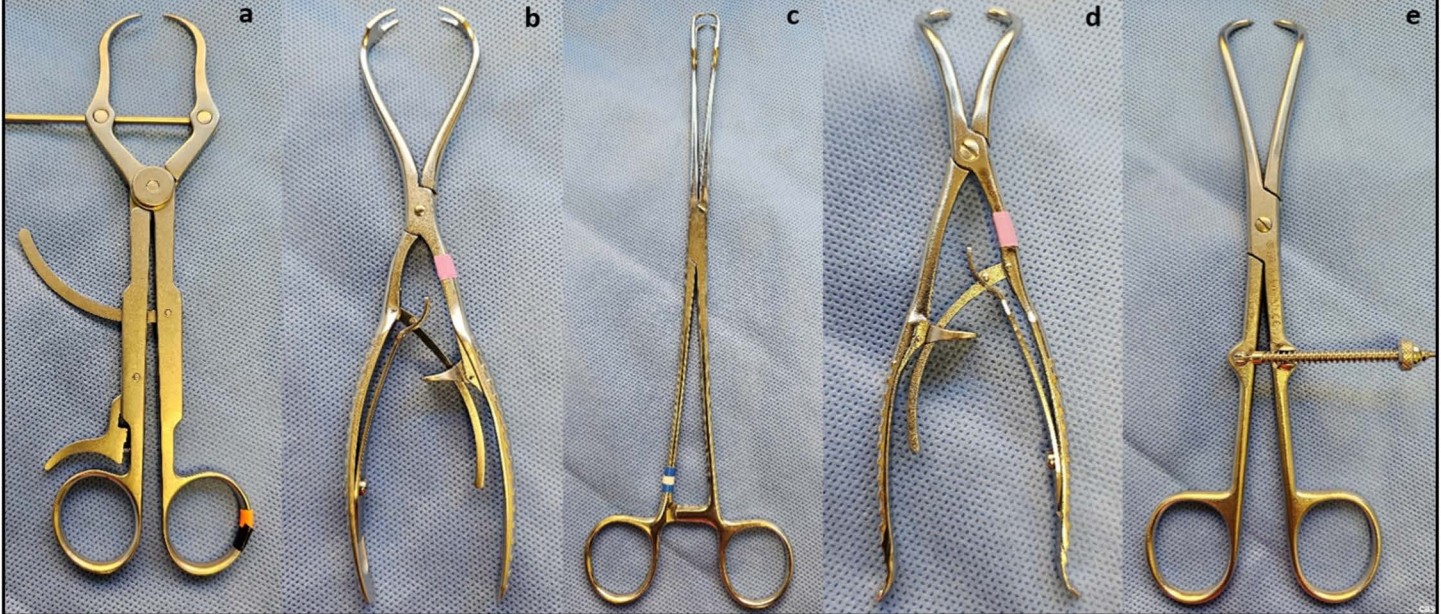

**Fig 3. This image shows the five bone reduction forceps used in this study:** Kyon FineTouch (a), Patellar (b), Vulsellum (c), point-to-point with soft-lock (d), and point-to-point with speed-lock (e).

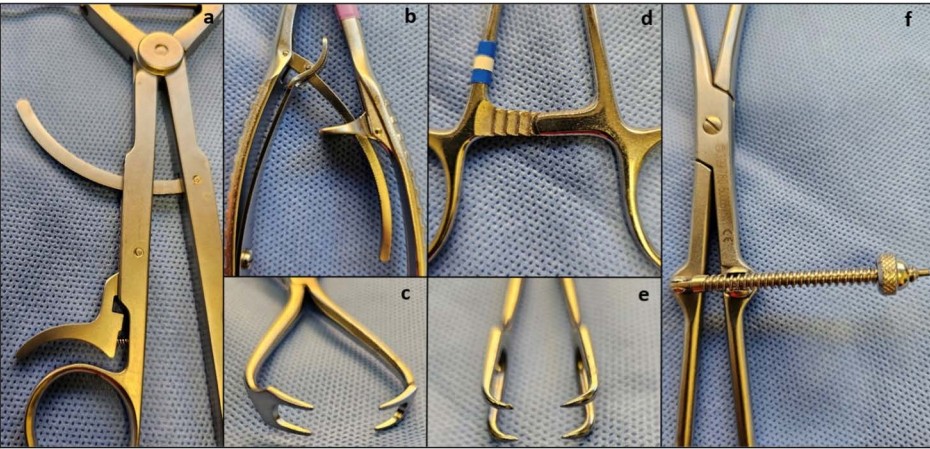

**Fig 4. This image shows the variable locking mechanisms and forcep tips between instruments used in this study: Kyon FineTouch** (a), soft-locking mechanism (b), Patellar forcep tips (c), ratchet-locking mechanism of the Vulsellum forceps (d), Vulsellum tips (e), and speed-locking mechanism.

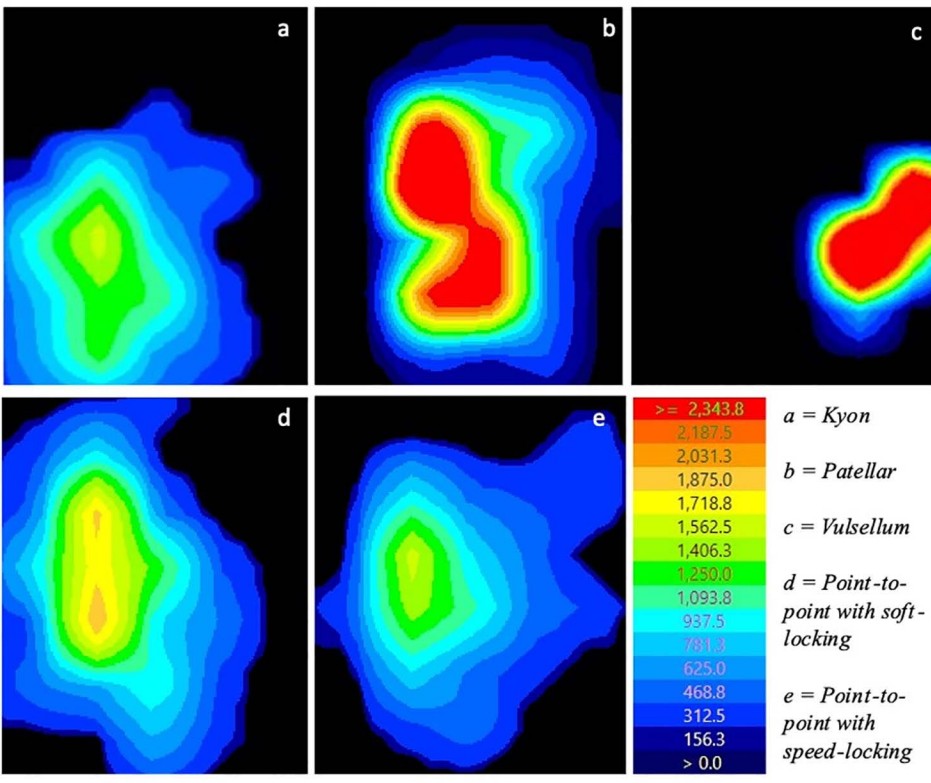

**Fig 5. Pressure maps at peak contact pressure for each forcep variation.**

compression were displayed as pixelated maps (Fig 5) showing variation in compression across the area. Interfragmentary pressure was plotted over the entire period of compression, allowing the final pressure plateau to be identified. Using the Tekscan program, the average area of compression (cm²), compressive force (kg), and peak pressure (kPa) were then determined for each trial at this plateau. These values were then recorded and compared.

## Statistical analysis

The data were tested for normality using the Shapiro-Wilk test and QQ plot. Because contact pressure and peak pressure violated the assumption of normality, they were taken natural log transformation. The mixed effect ANOVA model was used to determine a significant difference among five paired experimental groups using limb as a random blocking factor. Fisher's least significant difference (LSD) was used for the post hoc multiple paired comparisons. SAS version 9.4, release TS1M8 was used for all the analysis (SAS Institute Inc., Cary, NC, USA.), and a $P < 0.05$ was considered statistically significant.

## Results

No fragment fracture occurred during compression, thus allowing the inclusion of all humeri in the data set. If loss of reduction or forcep disengagement occurred, the data was excluded and the trial was then repeated until three data sets for each forcep were attained. Incidence of trial failure was not included as this could result from several variables (improper forcep placement, inadequate reduction, etc.).

Mean, median, standard deviation, minimum, and maximum values were listed by forcep type (Table 1). Area of compression did not vary significantly between the five groups ($P = 0.72$). However, there was a significant difference among the five groups in peak pressure ($P = 0.0003$). Patellar has the highest peak pressure, followed by Vulsellum. No significant difference in peak pressure was noted between Patellar and Vulsellum forceps.

The peak pressures of both the Patellar and Vulsellum forceps were significantly higher than point-to-point forceps with soft-lock ($P = 0.0011, 0.0116$), point-to-point forceps with speed lock ($P = 0.0001, 0.0017$), and Kyon FineTouch forceps ($P = 0.0008, 0.0084$). No statistically significant difference in peak pressure was noted among the lesser three groups.

## Discussion

In our study, we found that Patellar and Vulsellum forceps produced more interfragmentary compression compared to point-to-point forceps and Kyon FineTouch forceps. This led us to reject our first null hypothesis. However, we did not detect any statistically significant difference in the area of compression between the five groups.

All five bone reduction forceps generated some degree of compression across the fracture gap, which may be adequate for placement of fixation as described in recent literature [14]. The results of this study, however, show that use of either Patellar or Vulsellum forceps may promote greater compression of the fracture gap prior to introduction of either a lag or positional screw. The authors theorize that this may be secondary to tip configuration, as opposed to locking mechanism, due to the presence of two contact points on each side of the Patellar and Vulsellum forceps. This may allow for a greater distribution of forces on

**Table 1. Descriptive Statistics for Peak Pressure.**

| Method | Mean | Standard Deviation | Median | Minimum | Maximum |
|---|---|---|---|---|---|
| Kyon | 1131.36 | 372.59 | 1092.70 | 680.00 | 2217.40 |
| Patellar | 2210.31 | 885.23 | 2152.20 | 801.00 | 3856.60 |
| Point-to-point Soft Lock | 1164.30 | 382.48 | 1024.00 | 693.60 | 1946.40 |
| Point-to-point Speed Lock | 1005.94 | 247.44 | 1019.10 | 494.70 | 1338.10 |
| Vulsellum | 1904.83 | 751.65 | 1796.30 | 527.10 | 3268.80 |

each cortex during compression, in contrast with only one point of contact on the remaining models.

This construct also potentially reduces the risk of fragment fracture or damage to cortical bone at higher pressures as the force is distributed over two points of contact on each cortex. Additionally, the arrangement of the tips may allow for better purchase on each epicondyle, reducing the risk of forcep failure and reduction loss. Furthermore, Vulsellum forceps have a tip configuration that offers certain advantages such as the ability to drill between the tips. This allows for optimal placement of both the forceps and the screw at the middle of the fragment, resulting in maximum interfragmentary compression. The Synthes patellar forceps, in comparison, do not offer this advantage as the tips are situated too close to one another to facilitate drilling between them. However, as previously mentioned, there are currently no studies evaluating the clinical applications of different forceps. Therefore, any clinical extrapolation at this time is anecdotal.

Several studies have evaluated the biomechanical attributes generated by lag and positional screws. The aforementioned study evaluating interfragmentary compression generated by lag and positional screws in a porcine rib model also quantified the compression produced by both screw types when preloaded [14]. In this section of the study, the compression generated by the reduction forceps was paramount for reduction and compression of the bone fragments since the appositional screw would not generate any compression after placement. As a result, this study concluded that greater interfragmentary compression was generated by pre-loaded positional screws, as they tolerated placement with a higher torque (0.4 Nm) without stripping. The ability to apply fixation with a higher torque reduces the risk of screw pullout and provides a more stable fixation, demonstrating a benefit of positional screw use [13,16], and the importance of the compression force generated by the reduction forceps.

Another biomechanical study, in contrast, demonstrated the ability to generate greater interfragmentary compression with lag screws when placed in a similar manner [15]. This study yielded similar peak pressures for the lag screw group at time point 3 (mean 2043.18 kPa) when compared to peak pressures of Vulsellum (1904.83 kPa) and Patellar forceps (2210.31 kPa) in the current study [15]. This could indicate that use of pre-loaded positional screws may generate comparable interfragmentary compression to lag screws when specific reduction forceps are used. However, direct comparison between these studies cannot be performed due to the use of ovine humeri in the previous study. Additionally, this study also determined that lag screws produced a larger area of compression. The current study showed no significant difference between area of compression with various forcep types; however, clinical significance of this is unknown.

Several limitations of this study should be recognized. First, the population in this study was small and limited to cadaveric humeri of patients between 8 and 16 kg with no gross orthopedic abnormalities. Additionally, all humeri used in this study were acquired from adult patients. This limits the clinical extrapolation of this study as patients diagnosed with a lateral humeral condylar fracture are often skeletally immature or have been diagnosed with orthopedic disease (i.e., HIF), indicating that cortical and cancellous bone density may vary between in vivo patients and the population in this study.

Additionally, due to the *ex vivo* nature of the study and removal of soft tissue attachments, placement of the reduction forceps at the level of the medial and lateral epicondyles may be easier to achieve than in a clinical setting. Additionally, removal of soft tissues may decrease the risk of forcep failure due to the ability to place the tips directly on the bony surface.

Although the present study provided valuable information regarding the ability of certain bone forceps to generate greater interfragmentary compression, further investigation of various forcep types when placing both lag and positional screws is warranted. Additional studies

are necessary to determine if the difference in compression is clinically relevant following placement of either screw type. It is also unknown whether the data from this study can be extrapolated to other clinical settings where bone forceps are used to generate compression.

In conclusion, based on the simulated lateral humeral condylar fracture model, the use of Patellar and Vulsellum forceps generated a greater interfragmentary compression than either the point-to-points with soft or speed lock mechanisms or the Kyon FineTouch forceps. This study indicates that the use of these forceps as preloading mechanism for placement of either lag or positional screws may provide additional compression at the fracture site, thus achieving a more anatomical reduction.

## Acknowledgments

The authors would like to thank Xiaojuan Zhu, MS, PhD for her assistance in statistical analysis of the data in this report.

## Author contributions

**Conceptualization:** Madison Baskette, Cassio Ricardo Auada Ferrigno.

**Data curation:** Madison Baskette, Cassio Ricardo Auada Ferrigno.

**Formal analysis:** Madison Baskette.

**Investigation:** Madison Baskette, Cassio Ricardo Auada Ferrigno.

**Methodology:** Madison Baskette, Cassio Ricardo Auada Ferrigno.

**Project administration:** Madison Baskette.

**Resources:** Cassio Ricardo Auada Ferrigno.

**Software:** Cassio Ricardo Auada Ferrigno.

**Supervision:** Cassio Ricardo Auada Ferrigno.

**Validation:** Madison Baskette.

**Visualization:** Madison Baskette.

**Writing – original draft:** Madison Baskette.

**Writing – review & editing:** Madison Baskette, Cassio Ricardo Auada Ferrigno.

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
