## [Decision Letter · Decision Letter 0]

21 Oct 2024

PONE-D-24-34534Biomechanical Comparison of Various Bone Reduction Forceps in Interfragmentary Compression and Area of Compression in an Experimental Model of Lateral Humeral Condylar FracturesPLOS ONE

Dear Dr. Baskette,

Thank you for submitting your manuscript to PLOS ONE. After careful consideration, we feel that it has merit but does not fully meet PLOS ONE’s publication criteria as it currently stands. Therefore, we invite you to submit a revised version of the manuscript that addresses the points raised during the review process. Please make the revisions suggested by the reviewer to increase the clarity of your methods, results, and discussion.

We look forward to receiving your revised manuscript.

Kind regards,

Joshua William Giles, Ph.D.

Academic Editor

PLOS ONE

3. Please remove your figures from within your manuscript file, leaving only the individual TIFF/EPS image files, uploaded separately. These will be automatically included in the reviewers’ PDF.

Additional Editor Comments:

The authors have provided a clear description of their work, but a few areas should be improved based on the reviewer comments.

Reviewers' comments:

Reviewer's Responses to Questions

**Comments to the Author**

1. Is the manuscript technically sound, and do the data support the conclusions?

Reviewer #1: Partly

2. Has the statistical analysis been performed appropriately and rigorously? 

Reviewer #1: Yes

3. Have the authors made all data underlying the findings in their manuscript fully available?

Reviewer #1: No

4. Is the manuscript presented in an intelligible fashion and written in standard English?

Reviewer #1: Yes

5. Review Comments to the Author

Reviewer #1: Please state in title and in the abstract that the study is on CANINE lateral humeral condylar fractures.

Fracture Model paragraph, last sentence: The second osteotomy direction and the reasoning for this is unclear. The cut started at the epicondyle and was directed 45 deg disto-lateral to proxo-medial? A figure would be helpful.

Interfragmentary Compression paragraph: Please double-check your sensor is 4040, that sensor is not listed on Tekscan’s website. It is important to know due to resolution of the sensor.

Please describe the calibration procedure. Calibration of these sensors are key to accurate data.

Bone Reduction Forcep Groups: Somewhere you should spell out DPS to DePuy Synthes or reference Johnson & Johnson.

Data Collection:

“converted into a line graph” is not enough detail. Please describe if/how you defined the region of compression. Was the peak pressure the single “sensel” of the highest pressure, or was it the average pressure across the region and the peak of the average over the recorded time.

How was the position of the condyle repeatedly placed back into the original/best location for each compressive test? A mal-positioned (rotated) condyle would mean mating faces were not contacting and would change the pressure recorded. A non-repeatable positioning would also give large variability to “Area of Compression”. If you analyze repeatability of the 3 tests per forceps type, would give an indication of reliability of your technique and therefore forceps comparisons.

Please state whether or not the sensor extended beyond the mating faces of the fracture fragments. If not, how would you ensure the same region of the fragments contacted the sensor. Similarly, state if there is any “fragment” portion of the bone contacting parent bone that is not have sensing elements in-between. This would make a big difference, if there was load supported by mating faces but not being sensed.

How were the 3 repeat measures handled in statistical ANOVA?

In the methods, you state “The bone forceps were then engaged until the maximum interfragmentary compression prior to bone fragment fracture, forcep disengagement, or loss of reduction was achieved.”

There are no results given on whether any bone fractured, if so, how were the other forceps measured? There are no results given on if forceps stayed engaged or slipped of easily. That would be very useful information. Perhaps a high fleeting peak compression was made, but the reduction not able to be held, that is not useful clinically.

Discussion:

Sentence: “Based on the compression achieved by each group, all five types of bone reduction forceps are appropriate for holding the fracture in reduction during fixation.” There is no indication from your data that the pressures measured would hold during fixation. You would need a reference related to holding power, or perhaps to the other studies you mention later in the discussion.

Paragraph 5 (the comparison against reference 14): It is difficult to make any comparisons with this study to your study, given the lack of specifying in both studies if the measures are average pressure across the fragment, then the maximum of this data over time, or if the “peak pressure” or “interfragmentary compression” is the peak value (max pressure in any sensel at any time point). These are very different measurements.

Figure 4. Please state if the images are of the same bone. Are the images all at the same scale? They are all varying in shape. The reader cannot tell if shape is related to the bone contacting itself or it is simply a different bone.

6. PLOS authors have the option to publish the peer review history of their article (what does this mean? ). If published, this will include your full peer review and any attached files.

**Do you want your identity to be public for this peer review?** For information about this choice, including consent withdrawal, please see our Privacy Policy .

Reviewer #1: No

---

## [Author Response · Author response to Decision Letter 1]

17 Dec 2024

Thank you for all of your feedback. That was very helpful in realizing where the paper could be improved. The reviewer comments, as well as replies, have been attached to the submission as well as pasted below.

Journal Requirements:

1. All manuscript stylings have been adjusted to meet PLOS ONE’s style requirements. These include titling of the article and article sections, citations, and author information.

2. Data will be made available publicly.

3. Figures have been removed from the manuscript file.

4. All references are accurate and available when searching either by citation or DOI.

Reviewer Comments:

Reviewer comment: Please state in title and in the abstract that the study is on CANINE lateral humeral condylar fractures.

Reply: This has been adjusted.

Reviewer comment: Fracture Model paragraph, last sentence: The second osteotomy direction and the reasoning for this is unclear. The cut started at the epicondyle and was directed 45 deg disto-lateral to proxo-medial? A figure would be helpful.

Reply: The second osteotomy direction was performed in the usual direction of the fracture through the lateral ramus seen in patients with lateral humeral condylar fracture. This was added to the end of the sentence and a figure was added (Fig 1) following both osteotomies.

Reviewer comment: Interfragmentary Compression paragraph: Please double-check your sensor is 4040, that sensor is not listed on Tekscan’s website. It is important to know due to resolution of the sensor.

Reply: Thank you for noticing. The correct sensor is 4041. Here are the specifications: https://www.tekscan.com/products-solutions/pressure-mapping-sensors/4041

Reviewer comment: Please describe the calibration procedure. Calibration of these sensors are key to accurate data.

Reply: Calibration was achieved through placement of a 10 kg calibrated weight on the wire film sensor while resting on a level surface. The pressure sensor was then set to display pressure in kPa, as well as to show the maximum differences among the pixels measuring the pressure. This is now included in the interfragmentary compression section.

Reviewer comment: Bone Reduction Forcep Groups: Somewhere you should spell out DPS to DePuy Synthes or reference Johnson & Johnson.

Reply: Thank you, this has now been included in the bone reduction forcep groups section.

DATA COLLECTION

Reviewer comment: “converted into a line graph” is not enough detail. Please describe if/how you defined the region of compression. Was the peak pressure the single “sensel” of the highest pressure, or was it the average pressure across the region and the peak of the average over the recorded time.

Reply: The data was converted into a line graph by plotting pressure over time. This allowed us to identify where the pressure plateaued. The Tekscan software then calculates the average area of compression, compressive force, and peak pressure for this time period. This is now included in the data collection section (lines 129-132 in copy with tracked changes).

Reviewer comment: How was the position of the condyle repeatedly placed back into the original/best location for each compressive test? A mal-positioned (rotated) condyle would mean mating faces were not contacting and would change the pressure recorded. A non-repeatable positioning would also give large variability to “Area of Compression”. If you analyze repeatability of the 3 tests per forceps type, would give an indication of reliability of your technique and therefore forceps comparisons.

Reply: The fracture, having been artificially generated, allowed for uncomplicated reduction of the lateral condyle and precise interdigitation with the proximal humerus. This process was made easier by the presence of the condylar bridge and the excellent visualization of the joint anatomy, ensuring clarity and accuracy in the operation. The fracture fragment was anatomically reduced with digital manipulation prior to the placement of the bone reduction forceps to ensure that the measurements were consistent and accurate. This is now mentioned in the Interfragmentary Compression section. Additionally, after the compression was achieved, the contact area was verified to check for any decrease in size.

Reviewer comment: Please state whether or not the sensor extended beyond the mating faces of the fracture fragments. If not, how would you ensure the same region of the fragments contacted the sensor. Similarly, state if there is any “fragment” portion of the bone contacting parent bone that is not have sensing elements in-between. This would make a big difference, if there was load supported by mating faces but not being sensed.

Reply: The sensor was always oriented perpendicular to the interfragmentary interface. This allowed us to ensure there was at least a small amount of the sensor on each side that was in excess, ensuring the entire area of compression was included within the sensor. The complete surface of the condyle was fully covered by the sensor, ensuring that every inch of the surface was accurately mapped during the test. Additionally, utilizing a larger sensor than the tested area provided us with an extra “spare" sensor for greater reliability.

On the other hand, despite it being almost impossible to position the sensor perfectly during every testing cycle, it didn’t matter which part of the sensor was mapping the contact, as long as the intercondylar area was fully covering the sensor. This is because the full size of the sensor maintained the same quality in terms of density, resolution, and sensitivity. This means the captured data was not limited by the sensor position, but instead, represented the actual contacted force mapping

Reviewer comment: How were the 3 repeat measures handled in statistical ANOVA?

Reply: The three repeat measurements were included in the total data set, meaning that there were 21 sets of data for each forcep. The average peak pressure and contact area for each forcep were then compared using statistical ANOVA to evaluate for significant differences between the averages.

Reviewer comment: In the methods, you state, “The bone forceps were then engaged until the maximum interfragmentary compression prior to bone fragment fracture, forceps disengagement, or loss of reduction was achieved.”

There are no results given on whether any bone fractured, if so, how were the other forceps measured? There are no results given on if forceps stayed engaged or slipped of easily. That would be very useful information. Perhaps a high fleeting peak compression was made, but the reduction not able to be held, that is not useful clinically.

Reply: This is a good point. No bones were fractured during placement, which allowed us to include all of our data sets. Loss of reduction and/or forcep disengagement did occur; however, this was infrequent and could have resulted from user error, so this data was not included. This is now discussed in the results section (145 to 149 in the tracked changes copy).

DISCUSSION

Reviewer comment: Sentence: “Based on the compression achieved by each group, all five types of bone reduction forceps are appropriate for holding the fracture in reduction during fixation.” There is no indication from your data that the pressures measured would hold during fixation. You would need a reference related to holding power, or perhaps to the other studies you mention later in the discussion.

Reply: This sentence was poorly written and was intended to discuss the ability to hold the fragments in reduction, which was not clear. However, reference to one of the papers mentioned later in the discussion was added as recommended to tie in the discussion of compression, as well (lines 166-168 in marked copy).

Reviewer comment: Paragraph 5 (the comparison against reference 14): It is difficult to make any comparisons with this study to your study, given the lack of specifying in both studies if the measures are average pressure across the fragment, then the maximum of this data over time, or if the “peak pressure” or “interfragmentary compression” is the peak value (max pressure in any sensel at any time point). These are very different measurements.

Reply: This is valid. The Tekscan software, however, is used in both studies, meaning that peak pressure is the highest pressure generated in a single cell, allowing us to compare both studies. The idea behind using the peak pressure is that the pressure should be the greatest in the center of the fragment at the level of the ideal placement for the transcondylar screw.

Reviewer comment: Figure 4. Please state if the images are of the same bone. Are the images all at the same scale? They are all varying in shape. The reader cannot tell if shape is related to the bone contacting itself or it is simply a different bone.

Reply: The images are not all of the same bone, but the variance also occurs due to the use of different forceps and their placement. The scale is, however, the same in all images. This has been added to the description of the figure.

---

## [Editor Report · Decision Letter 1]

6 Jan 2025

Biomechanical comparison of various bone reduction forceps in interfragmentary compression and area of compression in an experimental model of canine lateral humeral condylar fractures

PONE-D-24-34534R1

Dear Dr. Baskette,

We’re pleased to inform you that your manuscript has been judged scientifically suitable for publication and will be formally accepted for publication once it meets all outstanding technical requirements.

Kind regards,

Joshua William Giles, Ph.D.

Academic Editor

PLOS ONE

Additional Editor Comments (optional):

Thank you for thoroughly addressing the reviewer comments.
---

## [Editor Report · Acceptance letter]

PONE-D-24-34534R1

PLOS ONE

Dear Dr. Baskette,

I'm pleased to inform you that your manuscript has been deemed suitable for publication in PLOS ONE. Congratulations! Your manuscript is now being handed over to our production team.

Kind regards,

on behalf of

Professor Joshua William Giles

Academic Editor

PLOS ONE